# Metal-Free Carbon-Based Supercapacitors—A Comprehensive Review

**Noureen Siraj \*****, Samantha Macchi, Brian Berry and Tito Viswanathan**

Department of Chemistry, University of Arkansas at Little Rock, Little Rock, AR 72204, USA; spmacchi@ualr.edu (S.M.); bcberry@ualr.edu (B.B.); txviswanatha@ualr.edu (T.V.)
**\*** Correspondence: nxsiraj@ualr.edu

**Abstract:** Herein, metal-free heteroatom doped carbon-based materials are being reviewed for supercapacitor and energy applications. Most of these low-cost materials considered are also derived from renewable resources. Various forms of carbon that have been employed for supercapacitor applications are described in detail, and advantages as well as disadvantages of each form are presented. Different methodologies that are being used to develop these materials are also discussed. To increase the specific capacitance, carbon-based materials are often doped with different elements. The role of doping elements on the performance of supercapacitors has been critically reviewed. It has been demonstrated that a higher content of doping elements significantly improves the supercapacitor behavior of carbon compounds. In order to attain a high percentage of elemental doping, precursors with variable ratios as well as simple modifications in the syntheses scheme have been employed. Significance of carbon-based materials doped with one and more than one heteroatom have also been presented. In addition to doping elements, other factors which play a key role in enhancing the specific capacitance values such as surface area, morphology, pore size electrolyte, and presence of functional groups on the surface of carbon-based supercapacitor materials have also been summarized.

**Keywords:** metal-free; supercapacitor; carbon electrodes; doping; capacitance; renewable

---

## 1. Introduction

Presently, fossil fuels are the primary resources consumed to meet current energy demands. For the last several years, energy demands have been increasing steadily due to an increase in the world population. As a result, fossils fuel reservoirs are depleting rapidly and the use of these resources also has a detrimental effect on the environment, such as global warming [1]. Due to the impending energy crisis, alternative and renewable resources must be used to design power generation and energy storage devices. In this regard, solar, wind, water, and geothermal resources are being considered for power generation. Batteries, fuel cells, and electrochemical supercapacitors have similarly gained much attention for effective energy generation and storage [2,3]. Among these, supercapacitors have gained considerable importance as energy storage devices for meeting many of the requirements for an alternate energy storage system.

Supercapacitors may also be referred to as ultracapacitors or double-layer capacitors. Fifty years ago, supercapacitors were recognized as the most promising approach to store energy, due to their outstanding characteristics such as high power density (100 times greater than conventional batteries [4]), long lifetime (lifecycle upwards of 10,000 cycles), and superior charge/discharge rate (up to 10–20 times faster than Lithium ion batteries). Due to their prompt charging response, supercapacitors can be used widely in digital cameras, automobiles, flashlights, elevators, portable media players, etc. Traditional capacitors cannot compete with supercapacitors since the capacitance of the latter is thousands of times higher than the traditional capacitor (μFarads vs. Farads) [5]. Supercapacitors can

be used independently or they can be combined with batteries or fuel cells for various energy storage applications [6].

To date, research progress in the arena of supercapacitors has involved seeking new inexpensive materials and economical, prompt as well as facile strategies to design supercapacitors with enhanced performance. Materials that have been exploited for supercapacitor fabrications can easily be divided into three main categories, viz. (1) carbon-based materials, (2) metal oxides, and (3) conducting polymers. Zhang group [7] presented a short summarized review for supercapacitor materials, and indicated advantages and disadvantages accompanying different materials. There is no doubt that metal oxides exhibit superior performance for supercapacitors applications (upwards of 1000 F/g). Therefore, metal oxides are very popular for supercapacitor applications owing to their superior performance among the above-mentioned materials. For that reason, Zhang et al. [7] has extensively covered metal oxide-based supercapacitors in the review, whereas carbon and conducting polymers materials were briefly discussed. Although metallic supercapacitors are known for their high capacitance, there are some drawbacks associated with them, such as their extremely high cost (e.g., Ruthenium (IV) oxide precursors priced around \$60 per gram), weak selectivity towards different species, and high toxicity towards the environment [8]. Whereas, the other two materials, carbon and polymers, are considered as environmentally safe compounds. However, the main disadvantage of conducting polymers is associated with their high instability due to their brittle nature [9]. Thus, an urgent need has arisen to restructure or even abandon current technologies and seek novel, green, and inexpensive materials. In this regard, carbon-based materials are gaining tremendous attention of researchers for supercapacitor applications [10], with activated carbons available commercially at about 15 USD per kilogram. The main focus of this review is to provide a detailed discussion about carbon-based supercapacitors developed in last 20 years, and novel strategies that can be utilized to improve the performance of carbon-based supercapacitors.

Various forms of carbon are highly sought after to obtain improved capacitance, such as carbon nanotubes, activated carbon, doped carbons, and many others. Though carbon materials usually exhibit lower capacitance than metal oxides and conducting polymers, their natural abundance in different forms and low cost make them an ideal candidate for supercapacitor applications. In addition, surface area and conductivity can easily be tuned in the natural existing forms of carbon (e.g., biomass). At present, extensive research is being performed using many different kinds carbon materials, and the main objective is to improve the performance of carbon-based supercapacitors. As a result of these studies, several examples of carbon-based materials have been made where supercapacitor performance has been significantly enhanced by various techniques such as heat treatment [11] or doping with heteroatoms [4]. Specifically, carbon nanomaterials have been demonstrated for similar or better supercapacitor properties than metal oxides [12]. A more detailed discussion regarding other strategies employed to enhance supercapacitor performance will be discussed later in this review, but are summarized in Figure 1. This review will substantially help readers to design new supercapacitors based on novel carbon materials with superior performance.

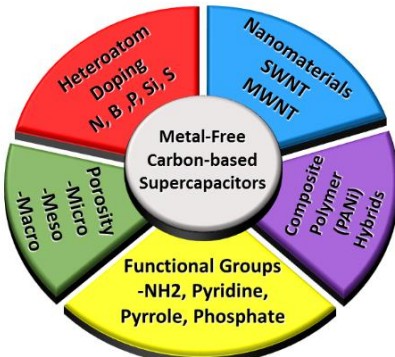

**Figure 1.** Strategies used to improve the performance of metal free carbon-based supercapacitor.

## 2. Electric Double-Layer Capacitor (EDLC)

Electrochemical capacitors can be divided generally into two classes: (1) Electric double-layer capacitors (EDLCs) and (2) pseudocapacitors. Normally, all carbon-based materials operate based on the EDLC principle. Therefore, only EDLC has been explained here, briefly. The mechanism of EDLCs is based on the storage of charge in thin double-layers, which is mainly present at the interface between the electrode surface and electrolyte solution. Since no charge transfer occurs between the electrode and the electrolyte, EDLCs are also known as non-Faradaic supercapacitors. Typically, all electrode surfaces possess EDLC behavior, and the capacitance recorded from this mechanism is much higher than the conventional dielectric capacitor. Ideal capacitors show a cyclic voltammogram of rectangular shape since the current is independent of potential. However, some potential dependent current is observed in EDLC due to high congestion of electrolyte at the double layer [13].

EDLC capacitance is represented by the following formula:

$$C = \frac{\varepsilon S}{d}$$

where $C$ is the capacitance, $\varepsilon$ is the relative permittivity, $S$ is the surface area of the electrode, and $d$ is the thickness of the double layer present at the interface between the electrode and the electrolyte. As can be deduced from the equation, EDLC is primarily dependent upon the surface area of an electrode material, as a high specific area provides a high surface for charge accumulation at electrode electrolyte interface. Generally, the capacitance result is proportional to the specific surface area for similar kinds of materials prepared using similar methods [14]. However, the surface area is not an all-encompassing parameter for determining material performance as the pore size distribution (micro-, meso-, and macropores) also plays a joint role [15]. Thus, electrode materials with suitable properties are very important to development of EDLCs.

With regards to supercapacitor application, some key characteristics of suitable material include large active surface area ($\geq$700 m$^2$/g for commercial ACs [4]), excellent conductivity, suitable pore size (for the electrolyte system chosen), surface wettability, high cycle stability, rapid charge/discharge rate, etc. These characteristics are highly desirable to design an efficient EDLC capacitor [16]. Therefore, extensive fundamental studies have been performed to investigate the properties of different existing forms of carbon materials. In this regard, carbon-based electrodes are commercially utilized as an ideal choice for EDLCs due to their physicochemical properties such as high conductivity and high surface area (1–2000 m$^2$/g), as well as high thermal and chemical stability. Furthermore, carbon materials exhibit extraordinarily high cycle stability as well as power density. Including all these properties, carbon-based materials are also highly attractive due to their significant low cost and abundant availability in nature [16]. Since carbon provides many advantages for applications such as EDLC, several designs of carbon have been developed for the purpose of energy storage capacity. Recent research of carbon materials for supercapacitor applications are classified as follows: activated carbon, doped carbon, functionalized carbon, carbon aerogels, graphene, carbon nanomaterials, and carbon quantum dots [17].

## 3. Carbon Materials

Carbon materials are of great importance for supercapacitor applications due to the presence of different forms of carbons and the fact that carbon is naturally abundant. Additionally, the chemical and thermal stability as well as good electrical conductivity of carbon are suitable characteristics to design low-priced supercapacitors. Furthermore, these materials can be made via environmentally friendly techniques such as pyrolysis [18] and hydrothermal treatment [19]. The final products possess a porous surface and exhibit high specific surface area.

### 3.1. Porous Activated Carbon

In order to attain high specific capacitance, porous materials that exhibit high surface area are promising candidates. Several methodologies have been employed to design porous carbon materials such as traditional chemical and physical activation and a combination of both processes, template-based methods, carbide-derived porous materials, etc. Fundamental physical characteristics such as specific surface area, electrical conductivity, pore size, shape, and their distribution are being studied in detail to determine the full potential of porous materials for supercapacitor applications.

Porous carbon materials synthesized by pyrolysis are vastly applied for industrial and laboratory applications. Biomass-derived carbon materials have acquired considerable attention in many fields, including supercapacitors [20–24]. Since most biomass sources are composed of carbon and oxygen, utilization of these materials for constructive applications provide a possible solution for global warming and other environmental challenges. Activated carbons are the least expensive to produce in comparison to other carbon materials. Their highly porous structure significantly enhances the surface area of the materials; thus, activated carbon has been employed as an electrode material for EDLC supercapacitor applications [16,25].

For supercapacitor applications, porous carbon materials can easily be obtained using chemical activation methods with various activating agents like potassium hydroxide (KOH), sulphuric acid ($H_2SO_4$), silver chloride (AgCl), zinc II chloride ($ZnCl_2$), and others. Amorphous carbon materials can be chemically activated using these agents, which aid in creating porous structures in carbon materials [26]. It is well known that an electrochemical activation method, normally used to activate carbon, can remarkably enhance the capacitance of final product in comparison to the parent carbon [27–29]. Chen and coworkers investigated the effect of KOH activation of AC performance and saw a ~185% increase in specific capacitance value post-treatment [30]. They attribute this enhancement to a high mesopore surface area (2505.6 $m^2$/g). Qin et al. achieved similar results using $ZnCl_2$ as an activating agent [31]. However, they attributed the high specific capacitance to a high specific surface area with significant micropore formation.

An outstanding approach to acquiring a high surface area for activated carbon and consequently, high specific capacitance, has been reported by Yushin and coworker [32]. A synthetic polymer named polypyrrole was used to synthesize activated carbon by performing a single activation step using KOH. Thermal treatment of polypyrrole produced a high percentage of carbonaceous residue, which led to a highly porous material with the highest specific surface area (greater than 3400 $m^2$/g) ever published for a carbon electrode. An exceptional specific capacitance of about 300 F/g was recorded in ionic liquid electrolyte for the activated carbon derived from polypyrrole. Among all other types of carbon-based electrodes, such as activated carbon, carbide derived carbon, carbon nanotube, and graphene, the activated carbon derived from polypyrrole exhibits the highest specific capacitance in ionic liquid electrolyte. Since 1-ethyl-3-methyl imidazolium tetrafluoroborate, an ionic liquid, was employed, an increase in capacitance was observed with increase in temperature due to a decrease in viscosity and an increase in conductivity of the ionic liquid.

In another report, porous activated carbon produced from waste tea leaves, a biomass source, was tested in an aqueous electrolyte [33]. The amorphous activated carbon displayed very high specific surface area with a value of 2841 $m^2$/g. An outstanding value of specific capacitance (330 F/g) was observed in KOH electrolyte with high cycle stability. About 92% of the initial capacitance was retained after 20,000 cycles, thus demonstrating high cycling stability. Similarly, another study demonstrated a sustainable synthesis of porous activated carbon using waste tree seeds [34]. A thermal pre-carbonization and subsequent activation with KOH yielded a material with 365 F/g capacitance value and retained 92% of that value over 5000 cycles.

### 3.2. Graphene

Graphene, which exists as an $sp^2$ hybridized carbon lattice, is well known for its remarkable electron delocalization characteristics. Therefore, it is widely used for numerous electrochemical

applications such as supercapacitors [35–39], next generation electronics [40], and sensors [41–45]. Herein, supercapacitor applications have been discussed in more detail.

Graphene is broadly used to fabricate supercapacitor devices due to their amazing mechanical strength, large surface area (2630 m$^2$/g), and excellent electrical properties such as high electrical and thermal conductivity, wide electrochemical window, and a high charge carrier mobility, ca. 20 m$^2$/V/s [46]. Due to these outstanding features, these materials have been exploited in different forms for energy storage devices. However, extraordinary characteristics of graphene showed specific capacitance results in various electrolytes (aqueous (135 F/g), organic (99 F/g), and ionic liquid (75 F/g) electrolyte), which are much lower than the expected theoretical capacitance value of 520 F/g [47,48]. These low capacitance values are ascribed to a decreased double layer formation. Therefore, it has been suggested that the theoretical capacitance value can only be attained when all active sites of the graphene electrode surface are accessible to electrolytes. In this regard, several methodologies have been utilized to design effective graphene-based materials for supercapacitor applications. The main goal of these studies is to prepare a graphene material with the maximum number of active sites. In these reports, researchers sought to enhance the specific surface area of graphene materials in addition to achieving tunable pore size with high electric conductivity [48–50]. Graphene prepared using different approaches have been tested in different electrolytes since the capacitance is dependent on electrolyte nature as well.

Methodologies employed to synthesize excellent graphene include epitaxial growth of graphene on substrate using chemical vapor deposition [51], exfoliation of graphite using AFM or in conventional organic solvent [52,53], gas phase synthesis of graphene platelets without any substrate [54], multilayered graphene synthesis [55], and many others [51]. Among these, exfoliation of graphite to graphene oxide is one of the most popular methods to fabricate supercapacitor electrodes based on graphene materials. This method is cost-effective and can be used easily for large scale commercialization. In addition, chemical modifications can easily be made on graphene oxide due to the availability of oxygen-containing functional groups. Furthermore, it allows easy tuning of the nanostructured size, while retaining the intrinsic specific surface area and electrical conductivity of the graphene.

Reducing agents employed to reduce graphene oxide directly impact the capacitance value of the resultant graphene materials. Reducing agents utilized at room temperature include hydrazine [56], dimethylhydrazine [57], hydrogen iodide [57], hydroquinone [58], sodium borohydride [59], etc. Besides their toxic nature, some of these reducing agents used for chemical reduction of graphene may introduce additional functional groups on the graphene surface during long processed reduction reactions.

Weak reducing agents cannot reduce all of the oxygen on the graphene oxide that enables easy penetration of aqueous electrolytes. Chen et al. [60] used hydrobromic acid, a weak reductant, to reduce graphene oxide. Hydrobromic acid was unable to reduce some stable oxygen groups present on graphene oxide. The presence of oxygen groups on graphene surface improves the wettability and facilitates the penetration of electrolyte into electrode pores. The presence of oxygen functionalities also yields pseudocapacitance and it was proven by observing the reduction signal of oxygen during cyclic voltammetry measurements. Hence, both EDLC and pseudocapacitance contribute to enhance the overall supercapacitor performance of the graphene. A maximum capacitance value of 348 F/g was observed using this graphene, where partial oxygen functionalities were reduced in 1 M aqueous H$_2$SO$_4$. Moreover, Chen et al. observed a continuous increase in capacitance value up until 2000 cycles. The 120% increase in capacitance from the initial value is attributed to the reduction of remaining oxygen during the cycling processes that improve the capacitor performance.

Recently, chemical reduction using various metal oxides in hydrochloric acid solution has been demonstrated as a more environmentally friendly approach to reduce graphene [61]. However, an electrode prepared by thermal and chemical reduction of graphene still suffers from small pore size, metallic impurity, and agglomeration [62]. Thus, it is necessary to design new methodologies

to restore the graphene material with less agglomeration and higher pore size. A highly conductive graphene material (1000–3000 S m$^{-1}$) was obtained by thermal nitridation of reduced graphene oxide [25]. The resulting materials are highly porous and exhibit high thermal stability. However, a significantly low value for the Brunauer-Emmett-Teller (BET) surface area (630.6 m$^2$/g) has been reported. These results demonstrate the great tendency of agglomeration of the nitrogen doped graphene material. Thus, small values for the specific capacitance are observed ca. 138.1 F/g. However, the capacitance performance can be tailored by altering the current density. Nevertheless, high electric conductivity, high porosity, and high connectivity of the nitrogen doped graphene sheets exhibit excellent stability. Moreover, these devices can be recharged within a minute.

Ruoff and coworkers prepared porous carbon with a BET surface area of up to 3100 square meters per gram. This was the highest surface area reported at the time of publication [63]. In this study, microwave-treated and thermally-exfoliated graphene oxide were chemically activated using KOH to enhance the surface area of carbon due to a high number of pores formed. This method can easily be scaled to an industrial level. These three dimensional sp$^2$ hybridized materials have pores with widths in the range of 0.6–5 nm size that exhibit high electrical conductivity, with low oxygen and hydrogen content. These supercapacitor electrodes have been tested with organic as well as with ionic liquid electrolytes and have been proven to show high values of gravimetric capacitance and energy density.

Graphene electrodes are also prepared by thermal reduction since it is a known green synthetic method [64,65]. Using thermal reduction, exfoliation of graphene was performed at a very high temperature to prepare a reduced graphene-based supercapacitor. The thermal reduction process is a comparatively expensive approach to reduce graphene because it requires high temperatures and a tediously long time. Furthermore, reduction at elevated temperatures generates carbon dioxide gas that can cause structural impairments on the graphene surface. Therefore, continuous research has been conducted to design new alternative approaches. To date, exfoliation reduction at room temperature is considered to be the best method to acquire high specific capacitance.

It is well established that graphene materials possess a high affinity to agglomerate themselves. Due to this restacking, pores on the graphene structure cannot be accessed by the electrolyte. Therefore, it significantly reduces surface area and minimizes the capacitance value. In order to solve this issue, Liu et al. [66] presented the curved morphology of a graphene sheet. The curved shape inhibits the graphene sheet's tendency to restack, and thus the electrolyte can easily reach into the pores, wet the graphene, and efficiently form an electric double layer. The graphene sheets with pore sizes in the range of 2–25 nm, were tested with ionic liquid electrolyte. The curved graphene sheets exhibited very high capacitance (100–250 F/g) with no contribution from pseudocapacitance. Typical flat shape graphene sheet examined under the same conditions with ionic liquid electrolyte, exhibited less than 10 F/g capacitance. This breakthrough research indicates the potential of graphene in energy storage devices. Hence, it is imperative to develop new strategies to expose the surface area of graphene, which will inhibit the restacking of graphene sheet. Thus, the electrolyte can access the exposed surface area and form a double layer, which consequently enhances the performance of the material for supercapacitor application.

### 3.3. Carbon Nanotubes

Carbon nanotubes (CNTs) are low-cost durable materials, which possess high surface areas. Carbon nanotubes have been explored for supercapacitor electrode applications due to their outstanding mechanical, chemical, electronic, and optical properties [6,67,68]. As a result of these interesting properties, these materials are being utilized in other applications as well, which influences the significant growth of commercial production of CNTs. In the literature, a number of papers and patents that describe CNTs are continuously increasing, as depicted in Figure 2 [68]. CNTs exhibit comparable capacitance values to activated carbon, although activated carbon possesses a larger surface area. The great performance using CNTs is ascribed to the utilization of maximum surface area of CNTs for continuous charge distribution [69]. Furthermore, their mesoporous characteristics allow

electrolyte to diffuse more easily that reduce the equivalent series resistances and thus improve the power output [70,71].

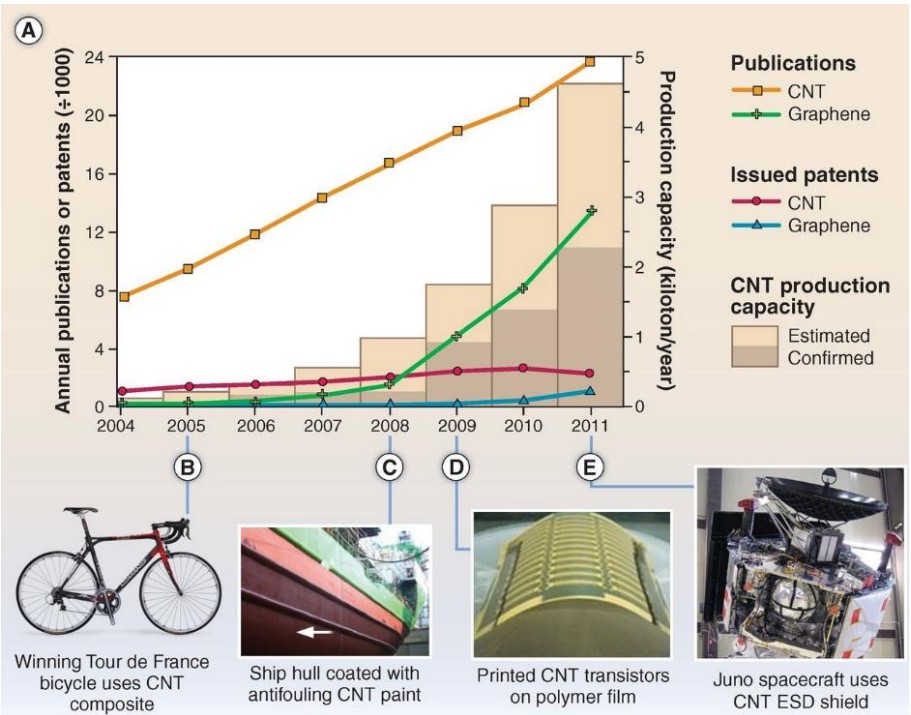

**Figure 2.** Relationship between CNT/graphene production and publications/patents [68].

Mainly, CNTs have been classified into two subclasses, viz., single wall carbon nanotubes (SWNTs) and multi wall carbon nanotubes (MWNTs). Highly conductive single walled and multi walled carbon nanotubes offer a large accessible pore surface area. In addition, the pore sizes of single wall and multiwall carbon nanotubes can easily be tuned for efficient diffusion of electrolyte ions. Due to this flexibility, both SWNTs and MWNTs have been employed for EDLC electrode application to acquire maximum power of the electrode [69]. Both SWNT and MWNT supercapacitor performance have been studied and capacitances are reported to be 180 F/g and 102 F/g for SWNT and MWNT, respectively [70,72]. Researchers are introducing new methodologies to use unbundled SWNTs to improve their performance. The important feature of a MWNT is the rapid discharge time of 7 ms for 10 MWNTs, up to 10 V. However, their low energy density due to their much smaller surface area limits their applications [6]. Rey-Raap et al. investigated the use of MWNTs as additive for biomass-derived activated carbons and found that at 2% CNT content the specific capacitance is increased from 139 F/g to 190 F/g [73].

Don Futaba et al. [74] presented a controlled fabrication method to acquire more highly dense and aligned SWNTs, which possesses high surface area. The performance of these newly synthesized SWNTs were compared with carbon solid electrodes and activated carbon. The results demonstrate a superior capacitance performance with a very small loss of capacitance as well as high power operation of SWNTs as compared to other two carbon electrodes mentioned earlier. This tremendous performance was attributed to ion diffusivity in porous aligned SWNTs.

*3.4. Carbon in Other Nanoforms*

Besides CNTs, carbon has also been used in different nanostructured forms for supercapacitor applications. Multilayer carbon spherical nanoparticles, also known as onion-like carbon (OLC) or carbon nano onions, have also been demonstrated for supercapacitor applications due to their high conductivities and dispersion characteristics [75]. These materials are important to fabricate

microsystem electrodes. There are several methods reported to design OLC electrodes. Among these, a very inexpensive method, which involves annealing nanodiamond powders, is widely employed to generate carbon onions. These materials are interesting for energy storage due to their specific surface area (which is comparable to activated carbon) and the accessibility of surface area for ion adsorption. Upon comparison with electrolytic supercapacitors, these micro supercapacitors exhibit four orders of magnitude higher capacitance, an order of magnitude higher energies per volume and comparable power per volume.

### 3.5. Carbon Aerogels

Carbon aerogels composed of carbon nanoparticle networks with interspaced mesopores display high conductivity that have a positive effect on electrochemical properties and as a result improves the power efficiency of the device. Normally, carbon aerogels are used in electrodes for EDLC supercapacitor applications [76]. One advantage is that the use of a binder material is not required for developing carbon aerogels due to the presence of a continuous network system. Aerogels are typically produced using a sol-gel method that involves polycondensation of resorcinol or similar molecules in the presence of a catalyst such as sodium carbonate ($Na_2CO_3$) [14]. The porosity of these aerogels can be tuned by varying precursor ratio. This process can be time-consuming and expensive; therefore, new studies have sought to use greener methods to produce carbon aerogels. Lee et al. [77] produced aerogels from spongy fruit biowaste (jackfruit and durian) with specific capacitance values up to 591 F/g at a current density of 1 A/g. These materials retained 96% of their capacitance over 10,000 cycles.

### 3.6. Carbon Quantum Dots

A new type of carbon material has been introduced in the last few years, called carbon quantum dots [78]. They are famous for their low toxicity, high solubility and easy functionalization, and hence have been exploited for many applications [79–84]. Some methodologies have been applied to design carbon quantum dots using different precursors such as carbon nanotubes (CNTs) [85], $C_{60}$ cages [86], graphite [85], graphene sheets [86], carbon fibers [82], polyethylene glycol, saccharides [87], etc. Well-defined geometrical structures have been developed from carbon quantum dots by controlling optical and electronic coupling characteristics.

Recently, these materials have become very popular for supercapacitor applications. These quantum dots based on carbon materials have been used as electrode materials [88] in solution and also with solid electrolyte [89]. Porous carbon quantum dots were synthesized by chemical treatment of $C_{60}$ with KOH at high temperatures. Dimensions of these carbon quantum dots have average height and lateral size of 1.14 and 7.48 nm, respectively. Electrochemical performance by the electrode using carbon quantum dots showed 106 F/g specific capacitance. After 4000 cycles, 87.5% retention of capacitance was observed.

Quantum dots are also often used in addition with other common carbon-based supercapacitor materials to enhance capacitive performance and longevity of the material. Deng et al. [90] prepared a flexible all solid-state supercapacitor with quantum dot decorated carbon spheres. The device displayed a maximum specific capacitance of 237.3 F/g and retained exceptional energy and power output after bending at 154°. Zhoa et al. [91] reinforced carbon nanofibers with graphene quantum dots and saw an increase in specific capacitance from 115 to 335 F/g. This is due to conductive network of crystallized quantum dots embedded in the nanofiber.

### 3.7. Doped Materials

A minimum amount of other elements administered into a material is known as doping. Structural and electronic distortion caused by invasion of heteroatoms leads to changes in the properties of carbon-based materials. Thermal stability, localized electronic state, band gap, fermi level, charge transport, and optical characteristics are some of the properties that are directly affected

after incorporation of any heteroatom into a carbon-based material [14,92,93]. Doping methodology is broadly employed to improve electrochemical properties of semiconductors. Such approaches have been exploited to improve the conductivity characteristics of carbon materials [94]. In this regard, different heteroatoms individually or multiple heteroatoms simultaneously were incorporated into carbon-based materials at lower concentrations to boost the electrochemical properties by introducing redox capable functional groups [95,96]. Both approaches have been proven to develop better supercapacitors in term of capacitance and stability.

### 3.7.1. Nitrogen Doping

Nitrogen is the neighbor of carbon in the periodic table with five electrons in the outermost shell. Electron rich nitrogen creates distinct property changes when substituted for carbon in a carbon lattice. The high electronegativity of nitrogen creates polarization in a carbon network, specifically in graphene which consequently influences the electronic, magnetic and optical properties of the material [97]. By using nitrogen as a doping material, both *n* type and *p* type doping effects were observed. In *n*-type material, nitrogen donates the electron since nitrogen is electron rich in this state. *n*-type effects exist in graphitic (quaternary) compounds, whereas *p*-type doping effects are observed when nitrogen is electron deficient and withdraws the electrons from carbon lattice. Examples of *p*-type effect are pyridinic and pyrrolic configurations.

Significant improvement in the electronic performance of a device was observed as a result of nitrogen doping. Nitrogen doping is relatively simple and illustrates enhanced supercapacitor characteristics due to the tunability of electron localization upon nitrogen doping. Therefore, extensive research has been performed with nitrogen doping in comparison to addition of other heteroatom in carbon [98]. To address current and upcoming energy challenges, nitrogen doping into carbon-based materials has become a well-established and powerful method. Therefore, much research in this area is already reported and new designs and methodologies are being investigated. Numerous reviews on nitrogen doping have been published due to its importance in this field [99–102].

A number of methods and precursors are reported for nitrogen doping in carbon-based materials. A few of the more important examples of nitrogen doped carbon materials are discussed here. Several methodologies such as the normal post treatment of carbon [103], chemical vapor deposition [104], hydrothermal carbonization [103], and thermal treatment [105] are reported as a possible route to substitute nitrogen into carbon. Many different nitrogen precursors have successfully been used. Urea [106,107], nitric acid [108], ammonia [109,110], melamine [111], nitrogen containing biomass [112,113] nitrogen rich ionic liquids [114,115], as well as supramolecular ionic liquids [116] are some of the examples of nitrogen precursors that have been used.

A simple plasma process used to dope nitrogen on graphene led to an increased ultracapacitance behavior in the device (280 F/g) [117]. Examination of scanning photoemission microscopy results revealed that the four times increase in capacitance values was attributed to the presence of nitrogen at basal planes. Similarly, nitrogen-containing microporous carbon prepared by using a zeolite-templated approach showed an increase in specific capacitance compared to the pure microporous carbon, whereas the specific surface area and pore size of carbon materials remained constant [118].

Porous carbon materials doped with nitrogen using both ex situ and in situ doping by pyrolysis have been reported to enhance the supercapacitance characteristics of carbon due to increase in conductivity by delocalization of electrons [119,120]. In this regard, nitrogen rich precursors and polymers such as melamine, urea, polyacrylamide, polyaniline, human hair etc. have been investigated for the synthesis of nitrogen-doped carbon [32,121,122]. Recently, yogurt which contains proteins as a source of nitrogen rich precursor has been employed as a nitrogen source for carbon materials [123]. Examination of spectra obtained by using X-ray photoelectron spectroscopy revealed the presence of a high percentage of nitrogen in the carbon material, which has been developed by pyrolysis of yogurt. A high capacitance value of 225 F/g at 2 A/g current density with a surface area of 1300 m$^2$/g

is significantly higher than most of the nitrogen doped carbon materials designed using different precursors for nitrogen.

Chen and coworkers [124] developed nitrogen doped porous carbon using a mixture of urea formaldehyde resin and calcium acetate. They presented evidence that the mass ratio of both precursors and the carbonization temperature are highly important parameters to control the pore structure and size, surface area, and capacitance characteristics. As carbonization temperature increases, the size of pores decreases, and surface area increases. As a result, the highest capacitance was recorded at a high temperature with long-term electrochemical stability (approx. 5000 cycles).

In another publication, ethylene diamine and carbon tetrachloride have been employed as nitrogen and carbon precursors, respectively [125]. In this method, high temperature (>800 °C) pyrolysis is required, and the resulting material has less than 10 wt % nitrogen content. To avoid this tedious process, ionic liquids have been introduced as nitrogen precursor [114]. Ionic liquids are molten ionic salts with melting points below 100 °C. Ionic liquids have received considerable attention around the globe due to their outstanding properties such as high thermal stability, low vapor pressure, extended electrochemical window, high conductivity, tunability, etc. [126]. Siraj and coworkers have reported ionic liquids and frozen ionic materials for a wide scope of applications [127–131]. Nitrogen doped carbon materials prepared using ionic liquids as precursor of nitrogen exhibit porous structure. These materials exhibit 210 F/g capacitance in 6 M KOH at 1 A/g current density with approximately 95% retention in capacitance after 1000 cycles. These results indicate that ionic liquids are promising materials to fabricate highly stable and superior performance supercapacitors. This study opens the door for one more class of material which can be used as precursor for different heteroatoms.

### 3.7.2. Boron Doping

After nitrogen, boron is the second-most important element used for doping material due to its unique characteristics. Boron acts as an electron acceptor and causes defects when doped in carbon due to its three valence electrons. Because of an uneven charge distribution, the electronic structures of the materials are affected and consequently have an impact on electrochemical properties [132]. Carbon electronic structures are altered oppositely in the presence of boron when compared to nitrogen doping. The low electronegativity of boron in comparison to carbon is a highly desirable characteristic because it causes lower binding energy between boron and carbon bonds as compared to a bond between two carbons. Indeed, it is relatively easy to dope boron in carbon-based material [133].

Numerous efforts have been made to incorporate boron in carbon materials for acquiring enhanced supercapacitor performance. Therefore, several approaches have been reported for boron doping such as substitution [134], laser ablation [135], hydrothermal treatment [136], chemical vapor deposition [137], etc. In this regard, boric acid ($H_3BO_3$) is commonly used as precursor of boron since other boron-containing compounds such as boron hydride and boron chloride suffer from instability and corrosive issues [138]. However, other forms of boron precursors have also been implemented such as Ammonium Pentaborate Hydrate ($NH_4B_5O_8$ $4H_2O$) [139].

Normally, pyrolysis of boric acid at 900 °C is performed to produce boric oxide vapors, which can easily substitute carbon [140]. The main challenge in boron doping is to achieve a higher concentration of boron in carbon materials. Earlier, Cheng and his group [141] were able to achieve a maximum of 0.6% composition of boron in mesoporous carbon when they used a variable mole ratio of boron and carbon precursors. However, even though the boron concentration is less, the material showed 1.6 times increase in specific capacitance in comparison to pristine carbon. A further comparison between nitrogen doped mesoporous carbon and boron doped mesoporous carbon revealed 11 times less concentration of boron in ordered mesoporous carbon (0.21 F/m$^2$) and also slightly less interfacial capacitance than nitrogen doped ordered mesoporous carbon (0.31 F/m$^2$). Thus, it is believed that boron has tremendous potential to enhance the electrochemical properties of carbon materials. Because of this significance, boron doping has been continued and a year later, a 1.9 times increase in capacitance of carbon materials has been recorded upon boron doping [142]. Later, Niu et al. [140] successfully

attained the highest contents of boron (4.8%) in graphene using three hours of prolonged pyrolysis at 900 °C. This material showed outstanding capacitance (172.5 F/g), which is 80% higher than the pristine graphene. This increase in capacitance is attributed to pseudocapacitance behavior due to the presence of boron on the surface of graphene. Moreover, capacitance measured after 5000 cycles displayed 96.5% value of initial capacitance, which also demonstrated the exceptional stability of the boron doped graphene material. Furthermore, superior wettability and rate capability were also observed. In conclusion, remarkable electrochemical properties are highly dependent on the contents of boron. Recently, a specific capacitance of 200 F/g with high cycle stability (4500 cycles) was acquired in boron doped graphene nanoplatelets. In this procedure, graphene oxide was chemically treated with borane tetrahydrofuran solution, which is not only acting as a reducing agent but also as boron precursor [143]. Thus, there is a great potential to use boron as doping material to develop a highly efficient supercapacitor.

### 3.7.3. Sulfur Doping

The similar electronegativities of carbon and sulfur produce negligible polarization when sulfur is doped in carbon. However, the large structure of sulfur can cause significant disruption in the carbon lattice upon doping. The bond length between sulfur and carbon is larger than a carbon-carbon bond. Recently, sulfur has been studied as a doping material to tune the electrochemical performance [144]. The inserted sulfur is predominantly in the disulfide form (C—S—C) because of the reducing nature of carbonization [140]. Typically, sulfur-doped carbons are produced via introduction of sulfur-rich precursors prior to carbonization. However, some post-treatment methods have been shown to produce exceptional sulfur-doped carbons.

In one study, a sulfur doped carbon monolith polymer, where the atomic concentration of sulfur was about 0.83%, showed variable sizes of porous structures (which includes micropores, mesopores as well as macropores). It has been noted that sulfur doping improved the surface area (>2400 $m^2$/g) and exhibited high specific capacitance (175 F/g) with a higher cycle stability (2000 cycles) than the parent material without sulfur. However, the mechanism to explain the change in electrochemical behavior is still under investigation [145]. In 2017, Hao and coworkers achieved the highest amount of sulfur doping to date (8.2 wt %) by utilizing $H_2SO_4$ and biomass carbon source [146]. The materials achieved a specific capacitance value of 364 F/g, which is highly exceptional for biomass-derived carbons. One year later, Elmouwahidi et al. [147] doped agricultural waste with thioglycolic acid to yield a carbon material with an impressive content of sulfur (25.0 wt %). This material exhibited 325 F/g capacitance with a maximum energy density of 37 Wh/kg. This energy density is much higher than that for currently available supercapacitors, which show maximum values around 10 Wh/kg.

Post-treatments of carbon can also yield sulfur-doped carbons with unique surface functional groups. Zhang et al. demonstrated that post-treatment of sulfur doped graphene yielded a unique oxidized S-species that showed at specific capacitance of 270 F/g with 101% capacitance retention after 10,000 cycles [148]. Lei and coworkers developed sulfur doped carbons by utilizing a post-processing procedure with thiourea as dopant. They found that sulfur content decreased with increasing pyrolysis temperature [149]. They attribute this to the degradation of heteroatom functional groups at high temperatures. The optimum temperature for these materials was shown to be 800 °C, which contained 1.10 wt % sulfur and exhibited specific capacitance of 210 F/g. Studies of sulfur doped carbon will likely continue to grow in popularity in this field.

### 3.7.4. Phosphorous Doping

Phosphorous is a relatively new element in comparison to other elements discussed above for doping carbon. Phosphorous belongs to the same group as nitrogen. Since it is larger than nitrogen, it can cause more structural distortions. However, because of the low electronegativity of phosphorus in comparison to both nitrogen and carbon elements, the polarity of phosphorous and carbon bond is completely opposite with a longer bond length than nitrogen carbon bond. Significant effect on

electrochemical performance can be observed upon phosphorus doping in carbon-based materials. Although, phosphorus can be a promising candidate as a doping material, there still are very few examples of phosphorus doping in the literature [150]. Theoretically it has been proved that graphene bandgap properties are positively affected by phosphorus doping as compared to sulfur doping in graphene using the same concentration [151]. From computational studies, it has also been deduced that doping of phosphorous is energetically favorable [151,152].

Puizy et al. [153–157] documented several reports in regard to phosphorous doping. Typically, phosphoric acid has been employed as an activation material, but detailed characterization exhibits a small percentage of phosphorous in the resulting activated carbon materials [158]. Doping percentage of phosphorous is mainly dependent on reaction conditions that favor phosphate groups becoming linked with carbon. It has been demonstrated that chemical treatment of carbon with phosphoric acid produced phosphorous doped materials and also created micropores in the materials, which is highly desirable for high wettability of electrodes by electrolyte [159]. Other phosphorus containing precursor materials have also been employed in phosphorus doping such as monosodium phosphate [160] and sodium hypophosphite [161].

In regard to supercapacitors, the first report on capacitance performance using phosphorus was published in 2009 [162]. In this study, three different precursors of carbon were chemically activated with phosphoric acid. A detailed characterization of the phosphorous doped carbon surface revealed that the pore size of the materials was between 6.5 and 8.3 Å. This is the most effective size for the formation of double layer, which is obtained after phosphoric acid treatment. In addition, the electrochemical measurement of phosphorous-doped carbon revealed an astonishing value of large electrochemical decomposition potential for water (1.5 V), which is beyond water's theoretical decomposition potential value (1.23 V). Moreover, the supercapacitance value of 220 F/g was recorded with a rectangular shaped cyclic voltammogram. Stability of phosphorus enriched carbon were tested for 15,000 cycles at different potentials (5000 cycles at 1 V, 1.2 V and 1.3 V). Statistical analysis was also conducted and the results obtained from this analysis corroborate that suitable surface area of pores with sizes between 6.5 and 8.3 Å and the phosphorous doping are both playing a major role in attaining high capacitance results from phosphorous doped carbon materials. The analysis also suggested that phosphorous functionalities are responsible for pseudocapacitance and thus enhance the capacitance value. However, the cyclic voltammogram did not represent any significant pseudocapacitance contribution. The mechanism of enhanced capacitance using phosphorus doping material is not fully understood. This interesting report got the attention of researchers working in this field. Since then, scientists are developing new phosphorous doped materials and trying to understand the high voltage aqueous media operation as well as role of phosphorous for efficient capacitance performance [163]. Phosphorous is relatively new but currently an emerging element for the doping of carbon.

Karthika et al. [164] doped phosphorus using the same precursors, i.e., phosphoric acid into exfoliated graphene oxide and reduced exfoliated graphene oxide. As observed earlier, phosphoric acid chemical treatment activates the sample by producing micropores and phosphorous-containing functional groups. Examination of results revealed that both phosphorous doped samples exhibit high capacitance in comparison to phosphorous-free samples. Among phosphorous doped sample, reduced exfoliated graphene oxide showed the highest capacitance (367 F/g) over exfoliated graphene oxide (245 F/g) with high cycle stability. The highest capacitance in the former sample is attributed to high pseudocapacitance contribution due to phosphorous doping and an improved conductive network in the material. Another group, Zhao et al. [165], developed phosphorus-doped carbon nanobowls using a simple evaporation method. The phosphorous-doped bowls exhibited high surface area (1342 $m^2$/g) and showed exceptional capacitance retention (~95% over 40,000 cycles. It is believed that phosphorus is the most promising doping material to design better performance supercapacitors, which are highly stable at high potential.

### 3.8. Co-Doped Materials

Co-doped refers to more than one heteroatom being doped simultaneously in carbon based materials to further improve the electrochemical performance. Numerous examples of co-doped materials are found in the literature. Different combinations such as boron and nitrogen, nitrogen and phosphorous [166,167], nitrogen and sulfur [168], boron and phosphorous [169], silicon and phosphorous [170], have been investigated as co-doped carbon materials. However, many more combinations are possible.

Porous carbon materials co-doped with boron and nitrogen was introduced to further improve the specific capacitance based on the synergetic effect [171]. In this study, the size of the pores was also tuned and its effect on supercapacitor behavior are discussed. Wettability characteristics of electrodes' materials improved due to the presence of heteroatoms, which significantly changed the polarity. Thus, ions can diffuse easily to the electrode and increase the specific capacitance. This increase in capacitance is also ascribed to pseudocapacitive effect caused by doping materials.

An environmental friendly, quick, economical, and efficient method has been reported for the synthesis of a co-doped material by Viswanathan and his group [172]. In this regard, highly thermally stable phosphorous and nitrogen co-doped materials were synthesized by microwave-assisted carbonization of aminated tannin. A very inexpensive renewable raw material, tannin, was exposed to ammonium hydroxide to produce aminated tannin. Aminated tannin was further treated with phosphoric acid and microwaved for a short period of time to produce nitrogen and phosphorus doped carbon. The specific capacitance and surface area were found to be 161 F/g and 433 $m^2$/g, respectively. We also used different precursors to produce doped nitrogen and phosphorous carbon using microwave technology to achieve materials with high specific capacitance values (182 F/g) [173,174]. It has been observed that microwave technique usually produces high surface area materials [167]. A scanning electron micrograph of phosphorous and nitrogen co-doped carbon is presented in Figure 3 [175].

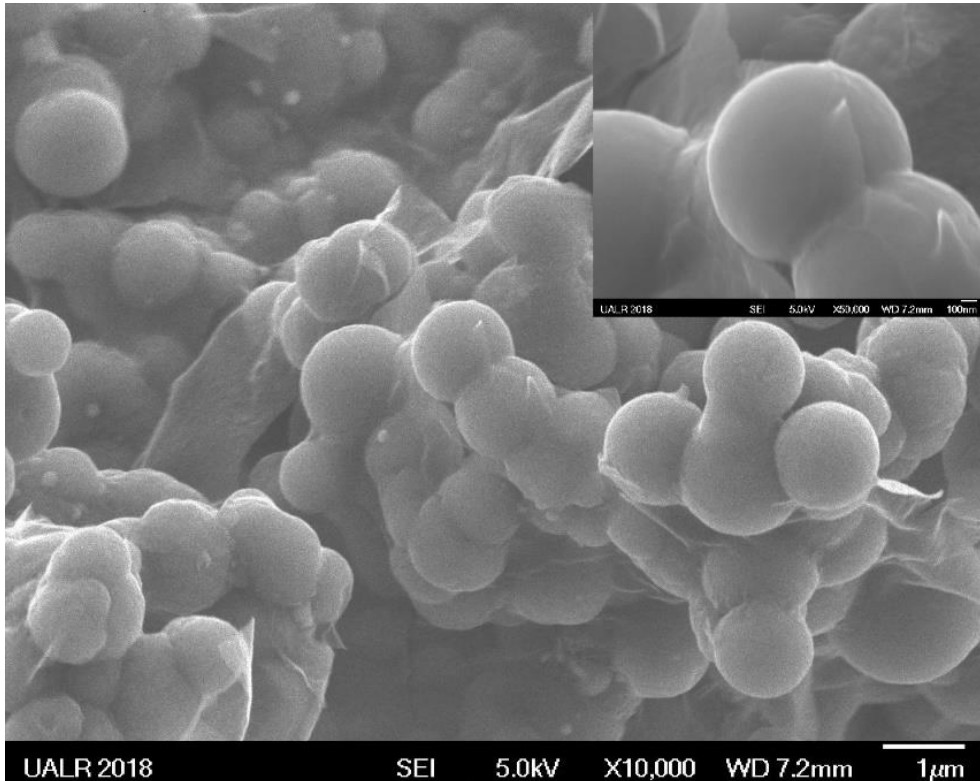

**Figure 3.** SEM image showing the spherical morphology of PNDC along with sheet-like structures. Inset shows the cross-section of some PNDC spheres.

Usually two heteroatoms are doped in carbon materials to form co-doped carbon materials. Recently, a report regarding three elements doped in carbon are presented, where silicon, nitrogen, and phosphorous are doped in carbon using a microwave-assisted method [176]. Three different precursors, thiamine, silicon fluid, and ammonium phosphate are mixed together. The presence of silicon, nitrogen, phosphate, as well as oxygen in carbon are confirmed using X-ray photoelectron spectroscopy. By varying the concentration of precursors, three different tri-doped materials were synthesized. The highest specific surface area obtained from tri-doped sample was 471 m$^2$/g. The tri-doped carbon exhibits the highest capacitance of 318 F/g in 6 M KOH, which depict tremendous stability after 2000 cycles. A cyclic voltammogram response observed in sulfuric acid is presented in Figure 4.

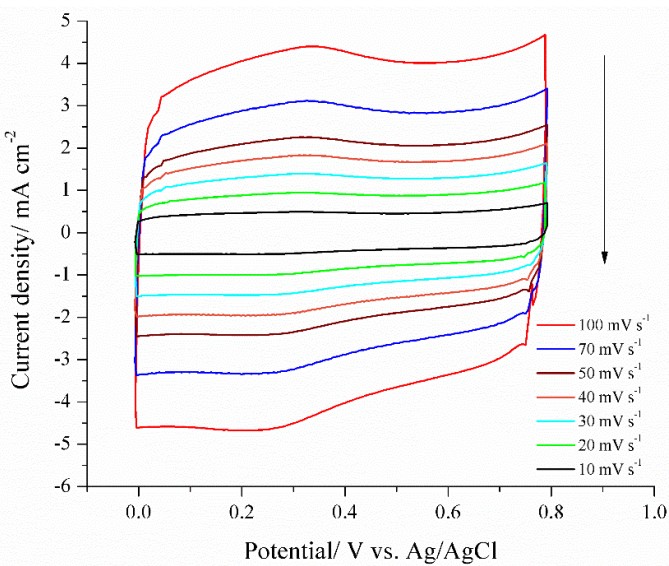

**Figure 4.** Cyclic voltammograms of NPSiDC-3 recorded at different scan rates in 1M H$_2$SO$_4$. Reprinted with permission from (S. K. Ramasahayam et al. ACS Sustain. Chem. Eng. 2015, 3, 2194–2202). Copyright 2015 American Chemical Society [176].

*3.9. Composite Materials*

Several materials are synthesized by incorporating transition metal oxides or polymers into carbon materials, which act as a new type of composite double layer faradaic supercapacitor [177–182]. These resulting materials are also recognized as composites since they are made up of two materials. These composite materials retain both electrochemical double layer capacitance (due to conductive carbon) as well as faradaic capacitance (due to the presence of metal oxides) features, which is also known as a hybrid capacitor. The dual combination merges the conductive property of carbon and the faradaic charge storage of metal oxides, exhibiting superior capacitor performance. As mentioned earlier, the main focus of this review is metal-free carbon-based supercapacitor materials. Thus, the dual capacitance strategy using metal oxides is beyond the scope of the current work. However, only few examples of polymer-based hybrid supercapacitor are discussed here.

Polyaniline (PANI) doped graphene composite exhibits tremendously high capacitance 480 F/g at 0.1 A/g, high conductivity as well as excellent cycling stability [183]. These excellent properties are observed due to the significant change in the composite structure due to the formation of homogenous composite material where PANI penetrated into the graphene or dispersed on the graphene surface. These materials are prepared using different mass ratios of graphene and PANI. Recently, Zhang and coworkers achieved 720 F/g (at 1 A/g) by polymerization of PANI onto MWCNTs using a β-cyclodextrin polymer. This composite also retained 97% capacitance over 5000 cycles [184].

High surface area and enhanced conductivity were also obtained by graphene doping in carbon to combine the characteristic of both materials [185]. Sucrose was used as carbon precursor but the

resulting material showed only 69 F/g capacitance. Several carbon-based quantum dots were also used in hybrid form for supercapacitor applications, which exhibit excellent cycling stability [186]. A hybrid material of graphene quantum dots and carbon nanotube exhibit 200% improvement in capacitance in comparison to CNTs free from graphene quantum dots [187].

## 4. Parameters Affecting Supercapacitor Performance

All the above carbon-based materials have been successfully demonstrated for supercapacitor applications and researchers are continuously exploring different approaches to enhance their electrochemical properties. This is a very active area for research, and numerous manuscripts and patents are being published continuously. In this section, some key areas are highlighted, which significantly affect the efficiency of supercapacitors.

### 4.1. Surface Area

Surface area is an essential parameter to forming an electrical double layer. In general, for similar types of materials, the larger the specific surface area, the higher the capacity that will be observed [14]. However, there is a limit to capacitance enhancement from surface area due to its relation to porosity. Nonetheless, several strategies have been employed in the literature to enhance the surface area of electrode materials such as thermal and steam treatment, chemical treatment (alkaline) and nitrogen flow rate, etc. [27,71,98,188]. In this regard, Lee group [71,72] exploited the effect of annealing temperature on the surface area of CNTs. Enhancement of smaller diameter pores was observed with an increase in temperature. Thus, by increasing the temperature, a large number of suitable pore sizes i.e., 30–50 Å was attained to increase the EDL capacitance [189]. The presence of ample pore diameters enhances the surface area and minimizes the resistance of CNT electrodes due to the diffusion of hydrated ions in pores. A maximum capacitance of 180 F/g was reported at the highest temperature (1000 °C), which was attributed to the increase in surface area, as depicted in Figure 5. However, another study of temperature effect on chemically treated activated carbon with KOH and HCl by Ruiz et al. [190] demonstrated lower surface area at higher temperature. This chemically modified electrode also showed carbon monoxide evolution and a decrease in specific surface area at higher temperatures resulting in a lower capacitance value. Sometimes, the direct correlation of specific surface area and capacitance has not been proven due to inaccessible micropores at electrode surface for electrolyte [191,192].

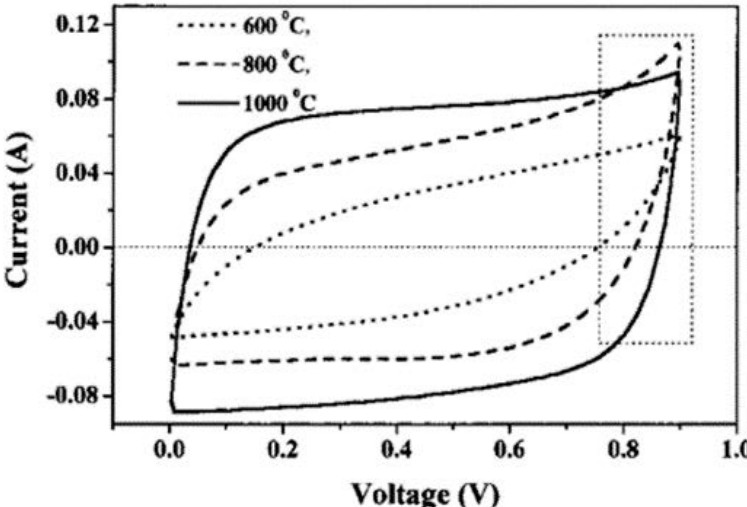

**Figure 5.** The cyclic voltammetric (CV) behaviors (sweep rate, 100 mV/s) for CNT electrodes at various heat-treatment temperatures [72].

## 4.2. Pore Size

The size and distribution of pores on electrode materials are of great importance for the absorption of electrolyte at the electrode-electrolyte interface. Diffusion of ions in the inner pores is highly important for supercapacitor applications. However, too high of porosity of electrode materials causes decrease in conductivity and consequently inhibits the maximum power density [69]. Furthermore, the size and texture of pores are important factors for capacitor performance [193,194]. Several methods have been developed to design porous carbon materials, which was reviewed by Kyotani [195]. The template carbonization is one of the most highly used approaches to design porous carbon materials of controlled and uniform pore size. Using this method, which was introduced by Kyotani, high surface area, high porosity and pores with similar sizes can be produced.

We have found a lot of discrepancy in the literature regarding the ideal pore size for supercapacitor applications; however, we have summarized some key findings in regard to pore size and distribution. It is known that an electrode does not possess a charge storage function if the size of electrode pores are smaller than the size of electrolyte ions [191]. Thus, the pore size of a material must be well suited for the electrolyte chosen (aqueous, organic, ionic liquid, etc.). In another study, the same group proved that larger pore sizes (mesopores) are vital for supercapacitor applications. Ordered mesoporous carbon (OMC) was first synthesized by Ryoo et al. [196], where the sizes of the mesopores can easily be tuned. These materials are highly interesting when determining the effect of ion diffusion in ordered mesopores. In this regard, Wang and coworkers [197] used 2-D ordered mesoporous carbons of two different pore sizes. Wang et al. reported that capacitance behavior increased as the pore size decreased when they used OMC of two different sized pores. The anomalous behavior of capacitance is explained as the presence of pore packaging defect, and materials without defect showed efficient ion transport and enhanced supercapacitor behavior.

Beguin group [118] also demonstrated the relationship between pore size and electrolyte size for supercapacitor behavior. In some studies, it has been mentioned that the optimal pore size of a particular electrode depends on the nature of electrolyte. They suggested that a pore size in the range of 4–7 Å is best for aqueous electrolytes whereas, for organic electrolytes, a slightly larger pore size of about 8 Å is optimal. From literature reports, we conclude that optimization of pore size is very essential in designing new materials for supercapacitor application. It is better to tune the pore size within the range of suitable pore sizes and optimize the capacitance properties in different electrolytes.

However, recent studies performed by Gogotsi and coworkers contradict the earlier finding and present a new idea. Gogotsi demonstrated that role of micropores which are less than 1 nm for non-aqueous electrolyte, provide the effective surface area for adsorption of ions and improve the capacitance [198–200]. This observation was confirmed by other scientists [201]. Huang et al. [202,203] explained these anomalous results by using a heuristic theoretical model and first-principles density functional theory (DFT) calculations. This model presents the correlation between the supercapacitor performance with pore size, surface area of electrode, and electrolyte concentration and size. These calculations were successfully proven for carbon-based supercapacitors in aprotic electrolyte and can be used for carbon supercapacitor with aqueous electrolyte. However, it cannot be applied to pseudocapacitors. This universal model considered the pore curvature instead of a parallel plate capacitor. Ions approaching pores mainly depends upon the size of those pores. In contrast to macropores and mesopores, ions located along the pore axis of micropores form an electric wire in a capacitor.

## 4.3. Functional Groups

An effective method to enhance the capacitance characteristics of carbon materials is to utilize appropriate pore sizes and functionalization on the electrode surface [118,119,204,205]. The introduction of heteroatom impurities, such as oxygen and nitrogen, increases the hydrophilicity of the carbon materials. Furthermore, the presence of functional groups on the surface enhances the wettability by adsorbing electrolyte ions and transporting them within the micropores of the electrode. In addition,

faradaic reactions due to the presence of functional groups results in faradic capacitance, which increases the total capacitance of the material. Thus, introducing certain functional groups on the surface produces an efficient capacitor.

Activated carbon surfaces can be obtained by oxidizing the surface groups or by treating activated carbon with different chemicals which possess functional groups [206–208]. Functional group formation is mainly dependent on the temperature. As an example, at high temperature, nitrogen functional groups appear where nitrogen is a part of aromatic ring with a delocalized charge or no charge, such as pyridine, pyrrole, aromatic amines, quaternary nitrogen and protonated pyridine. While at low temperatures, nitrogen is located outside the aromatic ring with a localized charge such as amides and protonated amides [209,210]. Functional groups of oxygen such as quinone, carbonyl, and carboxyl groups can enhance performance by facilitating ion diffusion at high current density. Quinone groups are especially favorable owing to their exceptional redox reactivity and electrochemical reversibility [211]. Because of the enhancement of electrode wettability, oxygen functional groups are not beneficial for organic electrolyte as they can block the pores [212].

Bandosz group [119,205] designed nitrogen and oxygen containing functionalized supercapacitor using coconut shell and wood. Melamine and urea were utilized to functionalize the surface of carbon electrodes. Examination of the surface revealed that nitrogen precursors and preexisting oxygen functionalities play a significant role in the development of functionalized activated carbon. Both nitrogen and oxygen-based functional groups such as pyridinic nitrogen, pyrrolic nitrogen, and quinone oxygen positioned in large pores of about 10 Å, exhibit pseudocapacitance in acidic electrolyte. In contrast, the smaller pores of about 5–6 Å showed EDLC. Furthermore, positively charged nitrogen functional groups displayed enhanced capacitance due to improved electron transfer at high current. Functionalized activated carbon derived from wood significantly improves the capacitance (300 F/g) in acidic media and exhibits high capacitance retention ratio (86%) at 1 A/g current load.

A tremendous increase in capacitance was observed when SWNTs functionalized with arylsulfonic acid were further treated with a conducting polymer pyrrole [98]. The functionalized composite electrode displayed an outstanding value of specific capacitance of about 350 F/g in basic media. This value of capacitance is found to be 7 times higher than the untreated buckypaper. It was discovered that macropores are highly responsible in enhancing the capacitance.

Chemical modification of graphene with detailed characterization was also reported and their supercapacitor behavior has been studied. A homogenous aqueous suspension of chemically modified graphene exhibited high conductivity [213]. Sahoo and coworker [214] studied the amine functionalization of graphene and demonstrated the electrochemical performance of its composite material with polyanailine (PANI). The composite material after graphene modification showed a reduced specific capacitance (193 F/g) in comparison to the specific capacitance obtained for graphene polyanailine composite (224 F/g). While there is a significant decrease in capacitance upon amine functionalization, thermal stability and cycling stability are significantly improved.

While many advancements have come in the area of functional group introduction into carbon based electrode materials, the commercialization of this process still remains a challenge. Zhang et al. recently discussed a scalable method to introduce nitrogen groups onto commercially available activated carbons [215]. They were able to achieve as high as 15 g in a one-step synthesis with 427 F/g in aqueous electrolyte. On the other hand, many researchers are avoiding functionalization completely and seeking out new methods of enhancing purely carbon materials like pristine graphenes [216,217].

## 4.4. Effect of Electrolyte

The electrolyte is another key component of supercapacitor design. It is electrically conductive and allows the transfer of ions between electrodes. The electrolyte directly effects the energy density and power density of a supercapacitor since both parameters are related to the cell voltage. In addition to high ionic concentration, high conductivity, an economical, highly stable electrolyte with wide potential

window is desirable to achieve better performance from the EDLC device [218,219]. The interaction between electrolyte and electrode depends on the surface morphology of the electrode as well as the size and structure of ionic species present in the electrolyte. Therefore, it is essential to choose the most optimal electrolyte to achieve high efficiency from an energy storage device fabricated using different materials [220].

A variety of different electrolytes, their reaction mechanisms, characteristics, and challenges in supercapacitors have recently been reviewed by Pal and coworker [221]. Pal et al. discussed different characteristics of electrolyte such as their thermal and electrochemical stability, ionic radius, viscosity, diffusion coefficient and conductivity, all of which influence the overall performance of supercapacitor. The most commonly used aqueous electrolytes exhibit higher capacitance values due to greater conductivity with smaller ionic radius, and low viscosity of concentrated solutions [222]. However, they suffer from a narrow electrochemical window, low cycling stability, and low energy density [3]. Therefore, organic electrolytes have been introduced due to an expanded electrochemical window; however, they possess bulky ionic specie with lower conductivity and high viscosity, which can negatively impact performance. Ionic liquid electrolytes exhibited promising characteristics such as low vapor pressure (green solvent), wide electrochemical window, and high thermal stability. However, they still suffer from high viscosity, low conductivity, and their high cost limits their large scale applications at an industrial level [223].

The use of redox active specie in the electrolyte can increase the supercapacitance due to pseudocapacitive contribution of the electrolyte. Pseudocapacitance can be introduced with the presence of heteroatom functional groups at the surface of electrode materials as well as the use of redox specie in the electrolyte. Taniki et al. [224] also presented that an ionic liquid electrolyte with redox reaction capability can improve the specific capacitance value of the supercapacitor. The details of other redox active electrolytes, solid state, and other electrolytes have been very well discussed in the recently published review article [221].

## 5. Fabrication Techniques

To achieve greater performance from carbon materials, numerous electrode deposition techniques have been introduced in the literature such as chemical vapor deposition, printing, spin coating drop casting, freeze drying, hydrothermal treatment, etc. [225]. The performance of graphene (one of the most promising carbon materials for industrial application) based electrodes has been improved significantly just by employing different fabrication techniques. The stacking tendency of graphene materials inhibits the charge/discharge processes due to the thickness of the electrode [226]. To avoid stacking, different template-based methods and chemical functionalization strategies have been implemented [227,228]. However, these methods are expensive and challenging to implement at an industrial level. Currently, toxic fluorine-based binders and hazardous solvents are being used to produce commercial supercapacitors, but many countries have restricted the use of poisonous chemicals for industrial production to a certain limit [229]. Therefore, environmentally friendly green solvents have been explored for the industrial fabrication of supercapacitors [230]. However, the preparation of paste or ink of the active electrode material in green solvent is challenging. To address these above-mentioned challenges, Garakani (energy storage materials) et al. [231] introduced a sprayable "green" ink for supercapacitor fabrication. The ink is composed of activated carbon and single/few layer graphene, which served as active material for supercapacitor. This spray technique highly scalable for industrial application and produced a highly homogenous layer which is easily accessible by the electrolyte. Garakani et al. also studied the performance of this electrode at different temperatures (−40/100 °C) in ionic liquid and organic solvent mixture-based electrolytes. There is still the need for a more optimized technique for commercial purposes.

## 6. Conclusions

Carbon materials have shown great potential in the supercapacitor field. Defects can easily be introduced in the carbon lattice by replacement with heteroatoms. Utilization of biomass-based renewable resource represents an attractive alternative for making doped carbon materials for supercapacitor application. Variations in structures that include different morphologies, dopants, composites, and functionalized materials have been evaluated in electrochemical capacitors. Recent research advancements on carbon-based materials indicate that toxic metals may be avoided in future supercapacitors without loss of efficiency. The main advantage in employing carbon materials is not only their low cost since different forms of environment friendly carbon exist abundantly in nature, but the easy tuning of electronic properties is possible by modifying porosity, dopant concentration, and functional groups. All strategies discussed in this review to enhance the supercapacitor performance have their advantages and limitations, and future researchers may keep these in mind while designing new methods and materials.

**Author Contributions:** Conceptualization, N.S., T.V. and S.M.; Methodology, N.S. and S.M. Formal Analysis, N.S. and S.M.; Investigation, N.S. and S.M. Resources, S.M. and N.S.; Data Curation, S.M.; Writing—Original Draft Preparation, N.S.; Writing—Review & Editing, N.S., S.M., B.B., and T.V.; Visualization, N.S. and T.V.; Supervision, N.S.; Project Administration, N.S., B.B., and T.V.; Funding Acquisition, S.M., B.B., and T.V. All authors have read and agreed to the published version of the manuscript.

**Funding:** This research was funded by a research and creative SEED grant and a signature experience grant from UA Little Rock.

**Acknowledgments:** The authors would like to acknowledge Iris Denmark for helping with the manuscript.

**Conflicts of Interest:** The authors declare that they have no conflict of interest.

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
