# Peer review of "Metal-Free Carbon-Based Supercapacitors—A Comprehensive Review"

_2673-3293, doi:10.3390/electrochem1040028_

Round 1
Reviewer 1 Report
Siraj et al. reported an overview of the use of metal-free (heteroatom doped) carbon materials for supercapacitors. The class of carbon materials are activated carbons, graphene, carbon nanotubes, carbon aerogels, carbon quantum dots, as well as their doped forms. Overall, the work is interesting and can give a general view on the topic of carbon-based supercapacitors. However, I would recommend some revisions, as listed here below:
1. For a scholar point-of-view, general statements should be more quantitative, providing accurate numbers in describing properties and performances, especially in the Introduction section. The authors use expressions such as “high power density”, “long lifetime”, “superior charge/discharge rate”. “capacitance is several orders of magnitude higher than”, “superior performance”, “high capacitance”, “high cost”, “weak selectivity”, “high toxicity”, “high instability”, etc… All these expressions should be supported by quantitative information/examples.
2. Industrial applications of active materials beyond activated carbons should be discussed to highlight the advent of alternative technologies to the commercially available ones, discussing both advantages and drawbacks. For example, some recent paper commented these aspects, especially for the case of graphene-based supercapacitors (M. Horn, et al., Current Opinion in Green and Sustainable Chemistry, 2019, 17, 42-48; M. Cossutta, Journal of Cleaner Production, 2020, 242, 118468; J. Tian et al., The Chemical Record, 19, 1256-1262.). In this context, market and cost analyses could be briefly commented for each class of materials. In fact, in supercapacitor field, the cost of the active materials represents a key-driver for design of practical supercapacitors. Even more, the scalability of each active material technology should be discussed, so that it can be correlated with targeted markets (niche or massive applications).
3. The techniques used to deposit carbon materials could be also discussed by analysing the resulting supercapacitor performances. In fact, the deposition technique of active carbon nanomaterials can determine the performances of carbon-based supercapacitors (for example, see M. A. Garakani et al., Energy Storage Materials, 2020, 34, 1-11).
4. In the Functional Groups section, the authors well discuss the advantages of chemically functionalized active materials. However, in commercial applications, the chemical functionalization approach is still rarely applied. Therefore, possible drawbacks should be also considered, while discussing how to overcome current limitations. Not by chance, several groups are currently focusing on exploiting the peculiar properties of pristine carbon nanomaterials, especially for the case of graphene and graphene-based hybrid electrodes (see for example: Journal of Power Sources, 2019, 437, 226899; Nanoscale Horizons, 2019, 4, 1077-1091, Nanoscale, 2019, 11, 17563-17570).
Reviewer 2 Report
COMMENTS FOR THE AUTHOR:
The reviewed manuscript describes metal free carbon based supercapacitors. In general, the experiments, results and their discussion included in the manuscript are presented in not convincing manner. I do, however, have some questions, major remarks and suggestions which should be considered in order to improve the manuscript prior its publication. They are listed below.
- Authors must present lines numeration in all documents to facilitate the reviewing of the manuscript.
- We detect a plagiarism in some part of the document . I suggest to rewrite some parts.
- All the manuscript must be reviewed and authors must include more references in all text . for example in the part introduction , authors presents 2 paragraphs without any reference.
- Authors give a series of carbon materials used for supercapacitor, starting from activated carbon, graphene , carbon nanotube ..etc, authors should present all type of materials in homogenised forms, that's mean, definition of material, preparation methods and modification methods for ameliorating capacitance .
- In different parts , authors need to introduce more details about the materials presented , for example . Last paragraph of the introduction, second paragraphs from the part 1.2 when they talk about the role of the surface area and carbon characteristics , last line when they talk about the advantages of EDLC. part 1.3. techniques used for the preparation of carbon materials....etc
- Page 7, authors talk about the chemical agents used for the activation of activated carbon and they give KOH but what about others agents used like AgCl, NaOH , H2SO4, ... etc
- Page 8. discussion not homogeneous, they present a different in formations but there is not relation between .
- Figure 2, page 14. Relationship between CNT/graphene production and publications/patents recopied without authorization. . DOI: 1126/science.1222453
- Page 16. Part 1.3.5 carbon aero gels. is not good presented and need more details like the others parts .
- Page 18. more reference should be introduced in the paragraph 2.
- Page 19. more details about doping methods used for the introduction of nitrogen.
- Page 21. Boron doping, More reference for the boron characteristics.
- Page 23. Please try to discuss the sulphur doping in the same methods like for nitrogen and boron. try to give more information's . and also try to give s recent a results for this parts. there is some new works published with good results. https://doi.org/10.1016/j.cej.2017.11.141.
- Page 25. No reference introduced in all this page . introduce more from a recent publication of phosphorus doped materials.
- 3.9. composite materials. try to give a more example of composites materials and theirs electrochemical performances and also more details on the mechanism of working composite materials. in general m try to present homogenous information's
- .1.4.1. surface area . authors say that larger the surface area, higher the capacity, this is not true . several works demonstrated that the increasing of the surface area had a limit . because the surface area is related to the porosity.
- figure 5. no reference and no authorization .
- also, in this part authors must discuss the effect of the surface area on the electrochemical performances. but not talking about the modifications methods.
- pore size. more references . more recent publications should be introduced and also recent results on the effect of the pore size. and its dependences with the electrolyte size.
- functional groups. authors should reorganized this part and also present detailed review of the effect of the functional groups on the electrochemical performances.
for example, oxygen effect , ...etc.
- authors should also discuss others parameters influencing the electrochemical performances such as, electrolyte type , the conductivity...etc, I suggest to introduce other paragraphs on the effect of the electrolyte.
- what about the effect of the metal doping. there is no discussion of its effect.
- I suggest also change of the title of the manuscript. because this review present no jut metal free carbon but others material types.
Round 2
Reviewer 2 Report
After revision of the new version of this paper, I accept it in this form
Just i have one remarque about the effect of the nitrogen and oxygen , authors say that The introduction of heteroatom impurities, such as oxygen and nitrogen, increases the hydrophilicity of the carbon materials.
i think this effect has limit . and we can detect this effect from some concentration. i think its better to be precise.
Good luck